# Patient Safety Culture in the Southern Region of Saudi Arabia: A Survey among Community Pharmacies

**DOI:** 10.3390/healthcare11101416

**Published:** 2023-05-12

**Authors:** Vigneshwaran Easwaran, Tahani Musleh Almeleebia, Mohammad Jaffar Sadiq Mantargi, Noohu Abdulla Khan, Sultan M. Alshahrani, Khalid Orayj, Osama Abdullh Amer Alshehri, Nawaf Yahya Hassan Alqasimi, Saad A. AlFlan

**Affiliations:** 1Department of Clinical Pharmacy, College of Pharmacy, King Khalid University, Abha 61421, Saudi Arabia; 2Department of Pharmacology, Batterjee Medical College, Jeddah 21442, Saudi Arabia; jaffar.sadiq@bmc.edu.sa

**Keywords:** patient safety, positive response rate, community pharmacy

## Abstract

Measuring patient safety culture in the community pharmacy can help with identifying areas for development. The current study is a descriptive, cross-sectional, electronic survey conducted among pharmacists working in community pharmacies located in the southern region of Saudi Arabia. The community pharmacy version of the “Pharmacy Survey on Patient Safety Culture” (PSOPSC) was used to collect data. The positive response rate (PRR) was calculated as per the guidance provided by the Agency for Healthcare Research and Quality (AHRQ). Based on the PRR, two least-achieved items (<25%) were taken for further analysis to identify the possible predictors. A sum of 195 pharmacists were included in this study and most of them were working in chain pharmacies. The highest PRRs were observed with teamwork (94.99), and patient counseling (94.13), followed by physical space and environment (93.07). The lowest PRRs were observed with staffing, work pressure, and pace (47.70), followed by communication openness (72.60). Specific characteristics, such as experience and the number of working hours, are significantly related to a poor PRR. The current study results indicate that the scope for improving patient safety exists in various areas of community pharmacies. However, it is necessary to prioritize the need based on a positive response rate.

## 1. Introduction

Patient safety is one of the critical pillars of healthcare systems around the world. Patient Safety is a healthcare discipline that emerged with the ongoing complexities of healthcare systems [1]. It has been reported by several researchers that pharmacy practice services can substantially improve patient safety and reduce hospital costs associated with medication errors [2,3].

The importance of patient safety is increasingly getting recognized worldwide nowadays [4]. A systemic review of patient safety culture in Arabic countries was performed and reported that it is important to promote the patient safety culture. Understanding the components and influencing factors of culture, and assessing the safety culture, is essential to developing strategies that create a culture committed to providing the safest possible care for patients [4,5]. The Kingdom of Saudi Arabia (KSA) has a well-established national-level medicine policy. This policy sets a direction for future development by focusing on institutional interconnection, improvement in cost-effective procurement and prescribing habits, a secure supply of good quality medicines, and the growth of the domestic pharmaceutical industry. It also establishes the patient’s safety by tracking the medicines throughout the supply chain using an electronic track-and-trace system [6]. In Saudi Arabia, community pharmacies are still product oriented. In 2018, the Ministry of Health (MoH) in Saudi Arabia regulated and restricted the supply of antibiotics without prescription. In addition, MoH also launched an initiative to get medicines to the public from private community pharmacies free of charge, instead of taking them only from government hospitals. However, this raises a concern about patient safety in community pharmacies [7]. Most of the studies conducted in Arab countries utilized hospital surveys on patient safety culture (HSOPSC) [8,9]. However, easy accessibility and the quality of pharmacy practice services provided by community pharmacies make this setting a suitable place for disseminating patient safety. Thus, it is important to understand the real-life situation regarding various domains of patient safety culture in community pharmacies. Moreover, measuring patient safety culture can help with identifying areas for development and understanding the changes in practice over time [8]. Patient safety culture in health care is usually influenced by multiple factors within the healthcare organization and helps with the prevention and reduction of errors [10]. Therefore, understanding the patient safety culture of community pharmacies will be helpful to improving the quality of the KSA healthcare program, by raising pharmacists’ awareness about patient safety issues and identifying areas of strengths as well as those that require improvement [11,12]. Numerous research studies are available in the context of patient safety from the perspective of various healthcare professionals, including pharmacists [13,14,15]. Despite the availability of abundant literature in this regard, there is still a scarcity of data concerning the assessment of patient safety culture in community pharmacy settings, particularly in the Kingdom of Saudi Arabia [16,17].

Hence, the current study was undertaken to analyze the patient safety culture and to identify the possible predictors and areas for improvement related to patient safety culture in community pharmacies located in the southern region of Saudi Arabia.

## 2. Materials and Methods

### 2.1. Study Design and Sampling

It was a descriptive, cross-sectional, survey-based study conducted among the pharmacists working in community pharmacies located in Abha, in the southern region of KSA. A non-probabilistic convenient sampling technique was used to recruit the study participants. All of the pharmacists working in community pharmacies located in this region were invited to participate in the survey, including student pharmacists, pharmacy technicians, and pharmacy assistants, even if more than one pharmacist was available in the same pharmacy.

### 2.2. Study Tool

The community pharmacy version of the “Pharmacy Survey on Patient Safety Culture” (PSOPSC) developed by the Agency for Healthcare Research and Quality (AHRQ) was used to collect data for the study [18]. It is a pre-validated, self-administered questionnaire that uses a 5-point Likert scale containing 40 items that measure 11 domains of patient safety culture. The domains included in the study are physical space and environment, teamwork, staff training and skills, communication openness, patient counseling, staffing work pressure and pace, communication about prescriptions across the shift, communication about mistakes, response to mistakes, organizational learning, continuous improvement, and overall perceptions.

### 2.3. Scoring and Positive Response Rate Calculation

The level of agreement for each item in the Likert scale ranged from strongly agree (5 points) to strongly disagree (1 point) for positively phrased items, and vice versa for negatively phrased items. The same condition was applied for a few items where the responses ranged from always (5 points) to never (1 point). The high scores (4 & 5) were perceived as positive responses and the scores of 1 and 2 were perceived as negative responses. The neutral response, missed response, and don’t know responses were excluded for positive response ratio calculation. The level of agreement for each item in the Likert scale ranged from 5 points to 1 point. The scores of 1 & 2 were perceived as negative responses. The positive response rate was calculated according to the guidelines provided in the PSOPSC by the AHRQ. The calculation procedure is to divide the frequency of (scores of 4 & 5) positive responses by the total number of responses, with the exclusion of neutral and don’t know responses.

Based on the positive response rate (PRR), the two least-achieved items were taken for further analysis to identify the possible sociodemographic predictors. The reliability was ensured by calculating the internal consistency using Cronbach’s alpha for the whole questionnaire and individual domains, and it was found to be good.

### 2.4. Data Collection

Both the English and Arabic languages were used in the questionnaire to conduct the survey. The research assistants met the pharmacists in person at their pharmacies at a time that was convenient for them. They were given an electronic device filled with a survey form (Google Forms), and responses were collected in the pharmacy itself.

### 2.5. Data and Analysis

Necessary statistical analyses were performed using the Statistical Package for social sciences (SPSS) version 22, for windows (IBM Corp., Armonk, NY, USA). The chi-square test was used to identify the predictors related to the items that received the least positive response rate. A *p*-value less than 0.05 was considered significant.

### 2.6. Ethical Considerations

The study procedure and protocol were approved by the ethical committee of King Khalid University. Ethical approval # is ECM#2020-1102. The informed consent form was included at the beginning of the electronic survey questionnaire and only those participants who gave informed consent were allowed further access to the questionnaire to participate in the study.

## 3. Results

A total of 206 pharmacists were invited to participate in the survey; 11 among them have disagreed to provide electronic consent to participate in the survey. A total of 195 pharmacists were included in this study and among them, the majority (80%) were working in chain pharmacies. Among those included in the study, 17.4% were pharmacy managers, 64.1% were pharmacists, 13.3% were student pharmacists and 5.1% were pharmacy technicians. Around 131 (67.2%) pharmacists were males. Of the pharmacists included in the current study, 63.1% completed their Bachelor of Pharmacy degree. Furthermore, 23.1% of pharmacists completed their PharmD degree. Only 4.6% have completed a Master’s degree in pharmacy. More than 50% of the pharmacists had less than 5 years of experience and 26.2% have experience of 6–10 years. Additionally, 6.7% of pharmacists have an experience of 11–15 years. The lowest percentage (4.1%) of pharmacists have an experience of 16–20 years. A total of 6.7% of the pharmacists have experience of more than 20 years. Among the community pharmacists included in the current study, 34.9% were handling more than 250 prescriptions per week and the remaining were handling less than 250 prescriptions. More than half of the community pharmacists were working more than 40 h per week. One-quarter of the community pharmacists were not familiar with their patients and around half of the population (53.8%) were somewhat familiar with their patients. Furthermore, 20.5% of the community pharmacists were extremely familiar with their patients (Table 1).

The positive response ratio was calculated for the individual items and derived for various dimensions of patient safety culture (Table 2). The highest PRRs were observed with teamwork (94.99), and patient counseling (94.13), followed by physical space and environment (93.07). The least PRRs were observed with staffing, work pressure, and pace (47.70), followed by communication openness (72.60). The PRR ranged from 47.70 to 95 across various patient safety dimensions.

Considering the individual questions, the highest PRR was observed with B2 (Pharmacists in this pharmacy encourage patients to talk about their medications, 96), followed by A9 (Staff works together as an effective team, 95.91), and A1 (This pharmacy is well organized, 94.99).

The lowest PRR was observed with B9 (We feel rushed when processing prescriptions, 14.67), and B16 (Interruptions/distractions in this pharmacy make it difficult for staff to work accurately, 19.21) These two items contributed heavily to poor patient safety culture. Hence, these two dimensions were further analyzed to estimate the involvement of any sociodemographic predictors.

The results revealed that some specific demographic characteristics are significantly associated with the poor positive response rate. The year of experience is significantly associated with the PRR of the pace of processing prescriptions (*p* = 0.023). Working hours is another predictor involved in affecting patient safety by modifying the item known as “we feel rushed when processing prescriptions” (*p* = 0.003). None of the other sociodemographic characters are significantly associated with the item “B9” (we feel rushed when processing prescriptions) (Table 3).

The analysis of item B16 (Interruptions/distractions in this pharmacy make it difficult for staff to work accurately) shows that the type of pharmacy (*p* = 0.019) and the year of experience of pharmacists (0.003) are significantly impacting patient safety. Other sociodemographic characteristics are not showing any statistical significance (Table 4).

Table 5 depicts the overall ratings with regard to the patient safety grade reported by the community pharmacies included in the study. It was observed that 49.7% & 29.7% of community pharmacies rated their pharmacy as excellent and very good, respectively. Additionally, 25% rated their pharmacy a lower patient safety grade. Only 25% of the community pharmacists reported positively that the overall patient safety grade is good or very good, or excellent.

## 4. Discussion

Most of the studies conducted in the past concerning patient safety culture focused on hospital settings. Very few studies have reported on community pharmacies. Our literature survey revealed that very few studies were conducted to report the trend of patient safety culture among community pharmacies in the Kingdom of Saudi Arabia.

Medication errors are the most frequent cause of a degraded patient safety culture in community pharmacies, according to research studies conducted all over the world [19,20]. However, it is equally important to identify the other possible reasons which may diminish patient safety. Therefore, it is essential to understand the concept of patient safety culture from the perspective of community pharmacists [1].

Using the PSOPSC, the current study investigated the patient safety culture from the community pharmacists’ perspectives. The response rate of the current study was 92.8%, which shows community pharmacists’ commitment to enhance patient safety; this percentage was found to be greater than that of prior studies in the field [3,21,22].

According to a study from Kuwait, the highest PRRs were observed for cooperation, organizational learning-continuous improvement, and patient counseling. These findings resemble our study almost exactly, where we found that teamwork and patient counseling had the highest PRRs [1].

The 36 elements of the questionnaire were added up to produce an overall mean score for patient safety, i.e., 82.32, which is similar to the Kuwait study (83.3) but higher than the studies from Malaysia and China [3,23]. The overall score about patient safety in the current study demonstrates that the community pharmacists from the southern part of Saudi Arabia are very well-aware of their responsibilities to improve patient safety. Teamwork showed the highest favorable response rate in our study. These findings are close to those of a study carried out among hospital pharmacists from Kuwait [24].

Similar findings were observed in hospital-based studies from Taiwan, Belgium, and the United States [25,26]. In addition to hospital pharmacists, studies conducted amongst community pharmacists also yielded a similar result to the current study [23,27].

Heavy workloads and inadequate staff contribute to mistakes in the pharmacy, which might lead to clinically significant problems [28,29]. In addition to our study, numerous domestic and international studies observed a lower PRR in staffing and work pressure. These findings suggest that pharmacists around the world feel the same way—that they do not have enough people to handle the workload, which has a direct impact on patient safety [1,3,21,23,26,30].

Communication openness among the pharmacy staff members within the pharmacy can help to prevent mistakes and improve patient safety [3]. Despite a higher PRR in communication openness, the mistakes were apparent. A lower PRR was observed in the domain “responses to mistake”, specifically in the item “Staff feel like their mistakes are held against them”. The staff members who committed mistakes usually had a variety of emotional distresses. Additionally, these results are in line with the study conducted in the United States [31]. The supportive culture within the work environment and constructive feedback may help those people to learn from mistakes [32].

The current study results indicated a positive approach by community pharmacists towards providing patient counseling and spending more time with the patients to explain the appropriate usage of medications. It was recommended by the World Health Organization that pharmacists must spend at least 3 min with the patients to provide patient counseling and orientation [33] and this tends to impact the patient outcome directly [34]. It is not surprising that the community pharmacists from our study showed a positive, and best, approach towards patient counseling because the role of the community pharmacist in patient counseling and appropriate patient counseling skills are taught in almost every pharmacy school in Saudi Arabia [35,36,37].

The study from central Saudi Arabia reported that the risk of dispensing error is continuously increasing [38], and the work environment is considered one of the great influencing factors of pharmacists’ work [16]. Pharmacy errors are associated with the lack of adequate space and unfavorable environment. The current study indicated that most of the community pharmacies in the study region were well organized, free from clutter, and supported by a good workflow. These results are reflective of various national and international studies [3,5,23,30].

Most of the community pharmacists included in the current study satisfactorily rated their pharmacy on patient safety. However, the PRR was lower in one of the items included in this domain, which demonstrates that a few pharmacies were emphasizing sales more than patient safety. A study from Malaysia reported comparable results to our study [23]. The overall grade given by the pharmacists on patient safety was excellent, which is in line with a similar study published by Alsaleh FM et al. [1]. The current study attempted to figure out the possible predictor for the lowest-scored items. The pharmacists working in standalone pharmacies, highly experienced pharmacists, and pharmacists with high familiarity with patients were processing the prescriptions hastily during peak hours. The pharmacists who were working in independent pharmacies and who have high experience felt that they had frequent interruptions in their pharmacy, which made it difficult to work accurately.

### Limitations

With a non-probabilistic convenient sampling method, the current study lacks generalizability; there might be the possibility for under or over-representation of the population. The current study is a self-reported questionnaire-based study that could be possibly influenced by response bias and socially desirable bias. Questionnaires are often completed voluntarily, leading to the risk of response bias, where only those who have had positive experiences may complete the questionnaire. This may interfere with the results and may provide a false impression regarding the patient safety culture in pharmacy practice and medicine. In addition, it is possible that respondents may provide answers that they feel are socially desirable, rather than truthful or reflective of their actual experiences. This can lead to distorted findings and inaccuracy in the results.

## 5. Conclusions

Pharmacists in the community must ensure medication safety throughout the medication use process. A strong culture of patient safety can help in appropriate utilization of resources. The current study shall provide basic evidence to design an appropriate future intervention and to direct available resources. Our study findings indicated that there was an overall positive perception among the community pharmacists in Abha concerning patient safety culture. It also pinpointed the strong points and potential areas that could use more development to enhance the culture of patient safety in neighborhood pharmacies. It reveals that the scope for improving patient safety exists among all of the domains. However, it is necessary to prioritize the need based on a positive response rate, preferably for the domain called staffing, work pressure, and pace. In addition, it is not a one-point end, but rather it requires continuous evaluation and monitoring to understand the change of practice due to time.

## Figures and Tables

**Table 1 healthcare-11-01416-t001:** Sociodemographic characteristics.

Characteristics	Number of Responses	Percentage
Pharmacy type		
Chain	162	83.1
Independent	33	16.9
Pharmacist position		
Managing Pharmacist	34	17.4
Pharmacist	125	64.1
Technicians	10	5.1
Student Pharmacist	26	13.3
Gender		
Male	131	67.2
Female	64	32.8
Last degree in pharmacy		
Bachelor	123	63.1
PharmD	45	23.1
Master’s	9	4.6
Others	18	9.2
Year of experience		
Less than 5 years	110	56.4
6–10 years	51	26.2
11–15 years	13	6.7
16–20 years	8	4.1
More than 20 years	13	6.7
Prescription volume per week		
Less than 250	127	65.1
More than 250	68	34.9
Familiarity with patients		
Unfamiliar	50	25.6
Somewhat familiar	105	53.8
Extremely Familiar	40	20.5
Working hours per week		
30–40	94	48.2
More than 40	101	51.8

**Table 2 healthcare-11-01416-t002:** Positive Response Rate (PRR) to patient safety culture dimensions [18].

Dimensions	Number of Positive Responses (Score of 5 & 4)	Number of Negative Responses (Score of 2 & 1)	Total Responses (Excluding Neutral and Don’t Know Responses)	PRR (PRR = Number of Positive Responses/Total Responses × 100)
Physical space and environment				93.07
A1. This pharmacy is well organized	164	8	172	95.35
A5. This pharmacy is free of clutter/untidiness	153	14	167	91.62
A7. The physical layout of this pharmacy supports good workflow	143	12	155	92.26
Team work				94.99
A2. The staff treat each other with respect	166	10	176	94.32
A4. Staff in this pharmacy clearly understand their roles & responsibilities	162	9	171	94.74
A9. Staff work together as an effective team	164	7	171	95.91
Staff training and skills				88.60
A3. Pharmacy assistants/helpers in this pharmacy receive the training they need to do their jobs	138	17	155	89.03
A6. Staff in this pharmacy have the skills they need to do their jobs well	151	14	165	91.52
A8. Staff who are new to this pharmacy receive adequate orientation	142	23	165	86.06
A10. Staff get enough training from this pharmacy	144	20	164	87.80
Communication openness				85.08
B1. Staff ideas and suggestions are valued in this pharmacy	119	35	154	77.27
B5. Staff feel comfortable asking questions when they are unsure about something	141	14	155	90.97
B10. It is easy for staff to speak up to their pharmacy manager (chief pharmacist) or pharmacy owner about patient safety concerns in this pharmacy	134	20	154	87.01
Patient counseling				94.13
B2. Pharmacists in this pharmacy encourage patients to talk about their medications	168	7	175	96.00
B7. Our pharmacists spend enough time talking to patients about how to use their medications	151	11	162	93.21
B11. Our pharmacists tell patients important information about their new prescriptions	150	11	161	93.17
Staffing, work pressure, and pace				47.70
B3. Staff take adequate breaks during their shifts	109	38	147	74.15
B9. We feel rushed when processing prescriptions	22	128	150	14.67
B12. We have enough staff to handle the workload	130	27	157	82.80
B16. Interruptions/distractions in this pharmacy (from phone calls, faxes, customers, etc.) make it difficult for staff to work accurately	29	122	151	19.21
Communication about prescriptions across shifts				86.84
B4. We have clear expectations about exchanging important prescription information across shifts	131	15	146	89.73
B6. We have standard procedures for communicating prescription information across shifts	134	26	160	83.75
B14. The status of problematic prescriptions is well communicated across shifts	141	21	162	87.04
Communication about mistakes				90.03
B8. Staff in this pharmacy discuss mistakes	132	19	151	87.42
B13. When patient safety issues occur in this pharmacy, staff discuss them	136	15	151	90.07
B15. In this pharmacy, we talk about ways to prevent mistakes from happening again	150	12	162	92.59
Responses to mistakes				73.82
C1. Staff are treated fairly when they make mistakes	155	10	165	93.94
C4. This pharmacy helps staff learn from their mistakes rather than punishing them	124	21	145	85.52
C7. We look at staff actions and the way we do things to understand why mistakes happen in this pharmacy	131	18	149	87.92
C8. Staff feel like their mistakes are held against them	43	111	154	27.92
Organizational learning—continuous improvement				90.94
C2. When a mistake happens, we try to figure out what problems in the work process led to the mistake	153	10	163	93.87
C5. When the same mistake keeps happening, we change the way we do things	135	17	152	88.82
C10. Mistakes have led to positive changes in this pharmacy	137	15	152	90.13
Overall perceptions of patient safety				72.60
C3. This pharmacy places more emphasis on sales than on patient safety	63	94	157	40.13
C6. This pharmacy is good at preventing mistakes	133	15	148	89.86
C9. The way we do things in this pharmacy reflects a strong focus on patient safety	144	20	164	87.80

**Table 3 healthcare-11-01416-t003:** Predictors of the lowest scored item (B9) under the dimension of Staffing, work pressure, and pace.

Characteristics	Number of Positive Responses	Number of other Responses	*p*-Values
Pharmacy type			
Chain	20	142	0.298
Independent	2	31	
Pharmacist position			
Managing Pharmacist	2	32	0.730
Pharmacist	16	109	
Technicians	1	9	
Student Pharmacist	3	23	
Gender			
Male	16	115	0.556
Female	6	58	
Last degree in pharmacy			
Bachelor	14	109	1.000
PharmD	5	40	
Master’s	1	8	
Others	2	16	
Year of experience			
Less than 5 years	13	97	0.023 *
6–10 years	7	43	
11–15 years	1	10	
16–20 years	0	8	
More than 20 years	1	12	
Prescription volume per week			
Less than 250	12	115	0.269
More than 250	10	58	
Familiarity with patients			
Unfamiliar	2	48	0.162
Somewhat familiar	14	91	
Extremely Familiar	6	34	
Working hours per week			
30–40	18	76	0.003 *
More than 40	4	97	

Chi square test, * <0.05 considered significant.

**Table 4 healthcare-11-01416-t004:** Predictors of lowest scored item (B16) under the dimension of Staffing, work pressure, and pace.

Characteristics	Number of Positive Responses	Number of other Responses	*p*-Values
Pharmacy type			
Chain	22	140	0.019 *
Independent	7	26	
Pharmacist position			
Managing Pharmacist	6	28	0.511
Pharmacist	20	105	
Technicians	0	10	
Student Pharmacist	3	23	
Gender			
Male	15	116	0.550
Female	14	50	
Last degree in pharmacy			
Bachelor	18	105	0.588
PharmD	8	37	
Master’s	0	9	
Others	3	15	
Year of experience			
Less than 5 years	13	97	0.003 *
6–10 years	7	44	
11–15 years	6	7	
16–20 years	3	5	
More than 20 years	0	13	
Prescription volume per week			
Less than 250	20	107	0.638
More than 250	9	59	
Familiarity with patients			
Unfamiliar	7	43	0.304
Somewhat familiar	13	92	
Extremely Familiar	9	31	
Working hours per week			
30–40	14	80	1.000
More than 40	15	86	

Chi square test, * <0.05 considered significant.

**Table 5 healthcare-11-01416-t005:** Overall patient safety grade in community pharmacy.

Overall Patient Safety Grade	Frequency	Percentage
Poor	1	5
Very good	8	4.1
Good	31	15.9
Very good	58	29.7
Excellent	97	49.7

## Data Availability

Data from this study are available upon request from the corresponding author.

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
