# Peer review of "Patient Safety Culture in the Southern Region of Saudi Arabia: A Survey among Community Pharmacies"

_healthcare, 2023, doi:10.3390/healthcare11101416_

Round 1

Reviewer 1 Report

1. The total number of participants is 195. In table 1 are included 31 males and 4 females. In the section gender and degree in pharmacy the total number of participants is not 195. The numbers could be corrected or explained.

2. The methodology of calculation of PRR should be explained. In table 2 it is not clear how 151 numbers of answers are 19 % in one situation and 90% in the other?

3. According to the authors there is implemented national medicine policy. What is its scope and impact compared with the examined in the study questions. 

Author Response

Point 1: . The total number of participants is 195. In table 1 are included 31 males and 4 females. In the section gender and degree in pharmacy the total number of participants is not 195. The numbers could be corrected or explained.

Response 1: It is a typo error. The first digit for all the said columns were missed. Rectified and updated in the revised manuscript.

Point 2: The methodology of calculation of PRR should be explained. In table 2 it is not clear how 151 numbers of answers are 19 % in one situation and 90% in the other?

Response 2: Please note that in table 2. We have not provided frequency of positive responses, we provided only the total responses excluding neutral and don’t know responses and positive response rate. However, for an easy understanding we added positive and negative responses also in table 2. in the revised manuscript.

Point 3: According to the authors there is implemented national medicine policy. What is its scope and impact compared with the examined in the study questions.

Response 3: The scope of national medicine policy and its scope and impact were elaborated further in the revised manuscript.

Reviewer 2 Report

I am pleased to be able to comment on this piece of research involving patient safety, as the topic is important. The Abstract is satisfactory and the Introduction fine. 

There are a number of issues with the questionnaire needing clarification. The sampling frame is unclear, as no indication is provided of how many pharmacists were eligible to respond, or whether more than one pharmacist could respond from the same pharmacy, Were individuals provided with a specific password? Was there a pre-notification email, or any follow-up emails? I assume the English version was used? Do all pharmacists in the KSA have adequate English language skills to understand the questions?

It reads to me that the two Questions B9 and B16 which if answered positively eg "most of the time" or "always"  would indicate low safety outcomes. As most respondents have indicated "never" or "rarely" that these are not major safety issues. These need to be re-evaluated and may need the whole paper to be reworked. based on the number of prescriptions per day the responses are likely to not elicit being rushed or interrupted.

Phrases in lines 108-111 need rewording.

Author Response

Point 1: The sampling frame is unclear, as no indication is provided of how many pharmacists were eligible to respond, or whether more than one pharmacist could respond from the same pharmacy, Were individuals provided with a specific password? Was there a pre-notification email, or any follow-up emails?

Response 1: We took non probabilistic convenience sampling technique for our study (Mentioned in methodology). All the community pharmacists can participate in the survey, even if more than one pharmacist is available in the pharmacy. – Revised in the manuscript

Since the survey was conducted in person, thus the questionnaire is not provided with any password. since the survey was conducted in person no notification or follow-up emails were sent.  

Point 2: I assume the English version was used? Do all pharmacists in the KSA have adequate English language skills to understand the questions?

Response 2: The survey was done bilingually by using both English and Arabic Language. Revised in the manuscript.

Point 3: It reads to me that the two Questions B9 and B16 which if answered positively eg "most of the time" or "always"  would indicate low safety outcomes. As most respondents have indicated "never" or "rarely" that these are not major safety issues. These need to be re-evaluated and may need the whole paper to be reworked.

Based on the number of prescriptions per day the responses are likely to not elicit being rushed or interrupted.

Response 3: Yes, you are right. If they answered most of the time or always would indicate low safety outcomes.

As per the guidelines provided by Agency for Healthcare Research and Quality (AHRQ), all the negatively phrased questions should be scored in a reverse manner. It is already mentioned in the methodology section of manuscript. Page 2, line number 91, 92 & 93.

As per guidelines, even in our study these questions were scored reversely, thus high scores were given for the responses never and rarely (5 & 4 respectively) and considered positive responses.

Therefore for the questions, B9 and B16 only few positive responses were recorded. (Few respondents indicated never or rarely which indicates lower patient safety outcomes)

Based on number of prescriptions it is probably possible that the pharmacists may not be rushing while processing prescriptions. However, we may need to consider other factors which interrupts the work. It is also possible that other factors may interrupt their work or processing of prescriptions. (eg. Phone calls, customers for OTC products or customer behaviour and etc.) In addition from the results of our study it shows that number of prescriptions are not significantly impacting the process of prescriptions.

Point 4: Phrases in lines 108-111 need rewording.

Response 4: Rephrased in the revised manuscript.

Reviewer 3 Report

This is an interesting topic with good support from other trials.  Some significant issues with description of materials and methods.  

Wording can be difficult.  The conclusion was accurate but the results were very difficult to understand with the descriptions of the data.  The discussion does not really discuss the importance of the results in relation to patient safety overall.  More data are provided then just safety data.  

The title indicates patient safety (more than that) and indicates pharmacists but it included techs and students.  

Wording (English) can be tough to comprehend at times.  Use terms that are not so technical.  For example, page 2 line 93 "on the spot"  Examples Page 2 51, "Plenty of studies"

Materials and Methods

How were people invited?   Then the response rate is calculated based on what--those who responded after invited.  How many were invited and did not participate.  

Scoring is not well described for positive versus negative?  What statistical methods used for internal consistency?  

Chi Square does not seem to be the appropriate test based on description of data?  Chi Square does not predict (it is observed minus expected).

Results discussion different than tables such as PharmD are included into Table #1 but not page 3 line 110

Tables hard to follow with text buried in between.  

Lines 126 to 138  hard to follow, not sure what this means?

Lines 143 to 148 hard to follow?

Lines 150 to 158  Is this based on C3 and C9?  Hard to tell

164 to 227 hard to follow

Conclusion is appropriate

Wording (English) can be tough to comprehend at times.  Use terms that are not so technical.  For example, page 2 line 93 "on the spot"  Examples Page 2 51, "Plenty of studies"

Author Response

Point 1: This is an interesting topic with good support from other trials.  Some significant issues with description of materials and methods. Wording can be difficult.

Response 1: Wordings are Revised

Point 2: The conclusion was accurate but the results were very difficult to understand with the descriptions of the data

Response 2: Description of the results are Revised

Point 3: The discussion does not really discuss the importance of the results in relation to patient safety overall.  More data are provided then just safety data.  

Response 3: Discussion part Revised

Point 4: The title indicates patient safety (more than that) and indicates pharmacists but it included techs and students.  

Response 4: Rephrased in the revised manuscript.

The title was revised

Point 5: Wording (English) can be tough to comprehend at times.  Use terms that are not so technical.  For example, page 2 line 93 "on the spot"  Examples Page 2 51, "Plenty of studies"

Response 5: Rephrased in the revised manuscript

Point 6: How were people invited?   Then the response rate is calculated based on what--those who responded after invited. 

Response 6: People were invited to participate in the survey by directly meeting them in person at their premises. - Mentioned in methodology section.

The positive response rate was calculated according to the guidelines provided by PSO-PSC by AHRQ – Updated in the revised manuscript.

Point 7: How many were invited and did not participate.  

Response 7: 206 subjects were invited and 11 among them were disagreed to provide electronic consent and to participate in the survey. - Updated in the revised manuscript.

Point 8: Scoring is not well described for positive versus negative?  What statistical methods used for internal consistency?  

Response 8: Scoring procedure was elaborated in the methodology section. Cronbach’s alpha was calculated - Updated in the revised manuscript.

Point 9: Chi Square does not seem to be the appropriate test based on description of data?  Chi Square does not predict (it is observed minus expected).

Response 9: Chi square test shall be used, when there is dichotomous type of data. Hence in our study the data included dichotomous type (frequency of positive responses and other responses) thus chi square test shall be appropriate.

Point 10: Results discussion different than tables such as PharmD are included into Table #1 but not page 3 line 110

Response 10: To avoid unnecessary duplication only higher and lower values were described in the description of the results. However as per suggestion it was included in the revised manuscript.

Point 11: Tables hard to follow with text buried in between. 

Response 11: Appropriate space provided

Point 12: Lines 126 to 138  hard to follow, not sure what this means?

Response 12: Sentences were rearranged in the revised manuscript

Point 13: Lines 143 to 148 hard to follow?

Response 13: Sentences were rewritten in the revised manuscript

Point 14: Lines 150 to 158  Is this based on C3 and C9?  Hard to tell

Response 14: Rewritten

Point 15: 164 to 227 hard to follow

Response 15: Rewritten

Round 2

Reviewer 2 Report

Thank you for revising the paper. It is clear that the sample is not a random sample, with the additional issue of several responses from the same pharmacy. The results should also show the number of individual pharmacies from which results were provided. In addition the limitations section in the Discussion section should be strengthened to address the sampling procedure adopted.

It is still confusing about the scoring of Questions B9 and B16. If the responses provided in the revised manuscript are as reported in the questionnaire then the scoring has not been reversed. It is unclear from the paper and the response to the reviewer's if or when the reversal has occurred.  One would expect the response heading are original results provided by respondents. In addition these responses if reversed before inclusion in the table are out of character with all other responses. This needs to be clarified, with only the raw responses provided in Table 2. The  heading of that table should be reviewed. These two findings have major implications for the remainder of the results. English grammar especially n the amendments needs attention.

Some of the English grammar especially written for the revised version needs attention.

Author Response

We authors would like to thank you for your sincere effort made for the betterment of our manuscript. We appreciate your comments and suggestions.

Point 1: It is clear that the sample is not a random sample, with the additional issue of several responses from the same pharmacy.

The results should also show the number of individual pharmacies from which results were provided.

Response 1: Please note that the responses are not from the same pharmacy, the responses are from various independent and chain pharmacies. 

Our study not concentrated on individual pharmacies. The analysis included only the total number of pharmacies regardless of the number of individual pharmacies.

Point 2: In addition, the limitations section in the Discussion section should be strengthened to address the sampling procedure adopted.

Response 2: As suggested, the limitation section was revised.

Point 3: It is still confusing about the scoring of Questions B9 and B16. If the responses provided in the revised manuscript are as reported in the questionnaire then the scoring has not been reversed. It is unclear from the paper and the response to the reviewer's if or when the reversal has occurred. One would expect the response heading are original results provided by respondents. In addition these responses if reversed before inclusion in the table are out of character with all other responses. This needs to be clarified, with only the raw responses provided in Table 2. The  heading of that table should be reviewed. These two findings have major implications for the remainder of the results.

Response 3: Please note that the reversing of the scoring for the negatively phrased items (Applicable for B9 & B16) was done at the start of the initial data analysis and mentioned in the first version of the manuscript. Thus, it was not revised or changed in the revised manuscript.

In addition, the headings of table 2 are revised for clear and easy understanding.

The raw data from the original questionnaire for the items B9 & B16 is given below:

B9: We feel rushed when processing prescriptions.

Responses

Never

Rarely

Total number of Positive responses

(Never + Rarely)

Sometime

Don’t know

Excluded

(Sometimes +

Don’t know)

Most of the time

Always

Total number of Negative

responses

(Most of the time + Always)

Number of pharmacists

8

14

22

32

13

45

63

65

128

Scoring for each respondent

5

4

3

0

2

1

B 16: Interruptions/distractions in this pharmacy (from phone calls, faxes, customers, etc.) make it difficult for staff to work accurately:

Responses

Never

Rarely

Total number of Positive responses

(Never + Rarely)

Sometime

Don’t know

Excluded

(Sometimes + Don’t know)

Most of the time

Always

Total number of Negative

responses

(Most of the time + Always)

Number of pharmacists

15

14

29

38

6

44

66

56

122

Scoring for each respondent

5

4

3

0

2

1

Point 4: Some of the English grammar especially written for the revised version needs attention.

Response 4: English grammar was checked for revised sentences, and re-revised wherever it is required.
